# Is Boccia XR an enjoyable and effective rehabilitation exercise for older adults?

**Masataka Kataoka**[ID][1]*, **Kyoji Sugiyama**[1], **Akira Iwata**[1], **Yumi Higuchi**[1], **Ryosuke Saga**[2], **Shinji Takahashi**[3], **Mitsuhiko Ikebuchi**[3], **Hiroaki Nakamura**[3]

1 Graduate School of Rehabilitation Science, Osaka Metropolitan University, Habikino, Japan, 2 Graduate School of Informatics, Osaka Metropolitan University, Habikino, Japan, 3 Graduate School of Medicine, Osaka Metropolitan University, Habikino, Japan

* kataokam@omu.ac.jp

## Abstract

### Background and objectives

Maintaining activities of daily living (ADL) in older adults requires muscle strength and physical activity. However, exercise motivation often declines with age. Enjoyable activities can enhance motivation and effectiveness. Boccia is a recreational sport with rehabilitation potential but requires substantial space. This study evaluated the enjoyment and lower limb muscle activity of "Boccia XR," a virtual adaptation designed for limited spaces, comparing it with traditional Boccia and treadmill walking.

### Methods

Eighteen healthy older adults (mean age 73.3 ± 5.4 years) participated. Each performed Boccia XR, traditional Boccia, and treadmill walking in random order. Mood changes were assessed using the Profile of Mood States 2nd Edition (POMS2), and lower-limb muscle activity was measured via electromyography (EMG).

### Results

Both Boccia XR and traditional Boccia significantly improved positive mood (Vigor-Activity) and reduced negative mood (Total Mood Disturbance) as compared to treadmill walking. Muscle activity analysis revealed that Boccia XR and traditional Boccia imposed muscle loads comparable to treadmill walking. Rectus femoris activity exceeded that of treadmill walking, and medial gastrocnemius activation was sufficient for strengthening in sedentary older adults during Boccia tasks.

### Conclusion

Boccia XR is an enjoyable and effective physical activity for older adults, requiring less space, than traditional Boccia while providing physical benefits similar to treadmill walking. It may enhance exercise adherence and overall function in space-limited settings.

**Data availability statement:** All relevant data are within the manuscript and its Supporting Information files.

**Funding:** The author(s) received no specific funding for this work.

**Competing interests:** The authors have declared that no competing interests exist.

## Introduction

Maintaining activities of daily living (ADL) in older adults sustained muscle strength and physical activity [1–3]. However, older adults often lose motivation to exercise, leading to decreased activity levels [4,5]. This decline is a global concern, as it contributes to decreased mobility, increased fall risk, and lower quality of life in older populations [6]. The World Health Organization (WHO) highlights the need for accessible and engaging physical activities to prevent frailty and disability in aging populations [7]. Psychological factors, such as social isolation and lack of enjoyment in conventional exercise, further hinder adherence to physical activity programs among older adults [6,8]. Since enjoyable activities can enhance motivation [9], it is essential to make exercise enjoyable, particularly for individuals with limited physical function. Enjoyment in physical activity is influenced by multiple factors, but prior research suggests that positive emotional states, such as increased vigor, are closely associated with perceived enjoyment [10]. Based on this, we assess enjoyment by evaluating positive mood changes during exercise.

Walking is a recommended exercise for maintaining physical function in older adults [11–15], and treadmill walking is commonly used as a substitute for ground walking in rehabilitation [16,17]. However, despite its benefits, treadmill walking is often perceived as monotonous due to limited environmental changes and minimal social interaction [16,18]. Therefore, exploring enjoyable alternative physical activities is essential for improving lower limb function in older adults.

We focused on "Boccia," an accessible and enjoyable sport that is safe and mood-enhancing for participants [19]. Purpose-driven tasks involving multiple participants have been shown to positively affect moods [20–23]. Upper limb exercises performed in a standing position can engage the postural muscles of the lower limbs and trunk, as well as the muscles of the ipsilateral leg muscle [24–27]. Since Boccia involves upper limb movements that activate the lower limbs and trunk muscle, it may serve as an effective form of lower limb exercise. While various sports, such as Tai Chi, aquatic therapy, and table tennis, have been used in rehabilitation, they often require specific environments, specialized equipment, or prior training. In contrast, Boccia is accessible to a wide range of participants, requires minimal equipment, and incorporates strategic elements that enhance engagement and cognitive stimulation. These characteristics make Boccia a promising rehabilitation activity for older adults. However, despite these advantages, Boccia is not widely implemented in rehabilitation settings due to its spatial requirements. Traditional Boccia courts require a significant amount of space (6.0 × 12.5 m), which is comparable to a badminton court [28], which limits their feasibility in hospitals and care facilities. This study addresses this limitation by exploring an alternative solution that enables Boccia to be played in more accessible setting.

To address this, we developed "Boccia XR" with 1-10 Inc. (Kyoto-city, Kyoto, Japan), enabling Boccia to be played in a smaller space using projected images. By integrating extended Reality (XR) technology, Boccia XR has the potential to enhance engagement and motivation in older adults, similar to the positive effects observed with Wii- and VR-based exercises [29–37]. Digital interventions are increasingly recognized as effective tools for promoting physical activity in aging populations, particularly among individuals who face barriers to conventional exercise [38]. Many VR-based exercise programs, such as Wii Fit and immersive VR training, require head-mounted displays or motion controllers, which may limit accessibility for older adults. Additionally, previous studies have reported that immersive VR experiences can induce motion sickness, particularly in prolonged use [39,40]. In contrast, Boccia XR eliminates concerns about motion sickness by not requiring a head-mounted display while preserving real-world physical interactions by allowing participants to throw actual balls, ensuring ease of use while incorporating digital engagement. However, the extent

to which Boccia XR provides comparable or superior benefits to conventional Boccia and treadmill walking remains unclear.

In addition to enhancing motivation, an effective exercise program for older adults should provide sufficient lower limb muscle activation to maintain or improve physical function. In this study, exercise load specifically refers to the level of lower limb muscle activity during each task, assessed using surface electromyography (EMG). Given that lower limb strength is critical for mobility and fall prevention in older adults, evaluating muscle activation patterns in Boccia XR is essential to determine its potential as a rehabilitation exercise. This study addresses this gap by examining the effects of Boccia XR on mood enhancement and lower limb muscle activity, comparing it with traditional Boccia and treadmill walking.

We hypothesized that (1) both Boccia XR and traditional Boccia would improve mood more than treadmill walking, and (2) lower limb muscle activity would be partially higher in Boccia XR and traditional Boccia compared to treadmill walking. To test these hypotheses, we analyzed mood changes before and after exercise and measured muscle activity during task execution to determine whether Boccia XR is an effective as traditional Boccia. A positive outcome would indicate that Boccia XR is beneficial exercise for older persons in limited spaces. This study aimed to evaluate the enjoyment quotient and exercise load of the Boccia XR, confirming its potential as a rehabilitation program for older adults.

## Materials and methods

### Participants

Eighteen healthy, community-dwelling older adults, aged 65–83 years (mean age 73.3 ± 5.4 years; 13 males and 5 females), participated in this study. They were recruited through the local Silver Human Resources Center (Habikino, Osaka, Japan), indicating their engagement in various daily activities, such as part-time work or community service, though not necessarily in structured or high-intensity exercise. Given their active lifestyle, they are representative of the general healthy older adult population. Participants met the following inclusion criteria: (1) aged 65 years or older, (2) able to walk independently without assistive devices, and (3) able to understand and follow instructions. Recruitment was conducted without incentives to minimize bias. While participants with prior experience in treadmill walking or Boccia were not excluded, none had previously played Boccia. A sample size calculation using G*Power software (Heinrich Heine University, Düsseldorf, Germany) determined that a minimum of 12 participants per condition was required. Further details on the power analysis are provided in the Statistical Analysis section.

Each participant completed three tasks over three days: (1) Boccia XR (XR), (2) traditional Boccia (BO), and (3) treadmill walking (TR). This study was approved by the Ethics Committee of Graduate School of Rehabilitation Science at Osaka Metropolitan University (approval number: 2022-129). Informed consent was obtained from all the participants.

### Interventions

Participants performed the XR, BO, and TR tasks at intervals of at least three days, with a maximum of two weeks to minimize potential carryover effects. While the minimum interval was set to three days to prevent residual effects from prior tasks, the actual washout period varied depending on participant availability and scheduling constraints. Block randomization was used to counterbalance task order. Participants were divided into three blocks of six, each beginning with a different task. Subsequent tasks were balanced within blocks to ensure even distribution.

In both XR and BO, each participant performed four sets of six throws following the standard rules of Boccia. Players throw six colored balls (red or blue) to place them as close as possible to a white target ball, the "jack". They continued throwing until their colored balls are closer to the jack than their opponent's. The side with more colored balls closer to the jack than the opponent scores points [28]. In both XR and BO, participants competed against an examiner who served as their opponent. The examiner maintained a consistent level of engagement across conditions to standardized gameplay interactions and minimize potential influence on participant performance. To ensure fairness and consistency, the examiner adjusted their level of play to match each participant's ability, preventing extreme skill differences from affecting game outcomes. This approach ensured that the competitive aspect remained engaging while minimizing variability in participant performance due to opponent strength.

**Boccia XR.** Boccia XR is a virtual adaptation of boccia that requires one-sixth the space needed for traditional Boccia (Fig 1). It is played by throwing actual balls at a projected image of a Boccia court displayed on a wall (Fig 2). The direction and distance of the ball are determined based on its speed as it passes between two sensors positioned 40 cm apart near the wall. Each sensor is connected to a PC via a Wi-Fi router using LAN cables. A notebook PC with the application installed is connected to a projector (ViewLight ME403U, NEC, Tokyo, Japan) via an HDMI cable, projecting the Boccia court onto the wall. To avoid interference with the throw, the projector is positioned diagonally to the participant's right, and keystone correction is applied. The participant throws the ball from the front of the projected image, ensuring accurate gameplay tracking. Calibration is performed by setting the center point of the sensor on the wall as the origin of the spatial coordinates ($x = 0$, $y = 0$) and inputting the coordinate positions of the four corners of the projected image (relative to the origin). The Boccia XR software automatically determines the throwing order and provides on-screen instructions.

Boccia XR does not currently incorporate sound or vibration feedback but provides immediate visual feedback by updating the projected image in real time. The system highlights the current throwing order and displays an indicator showing the position of the latest

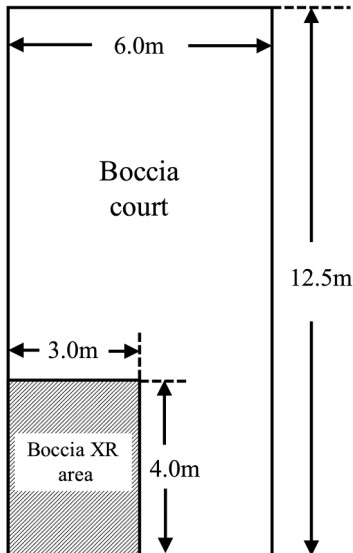

**Fig 1. Comparison of the sizes of Boccia XR and Boccia court.**

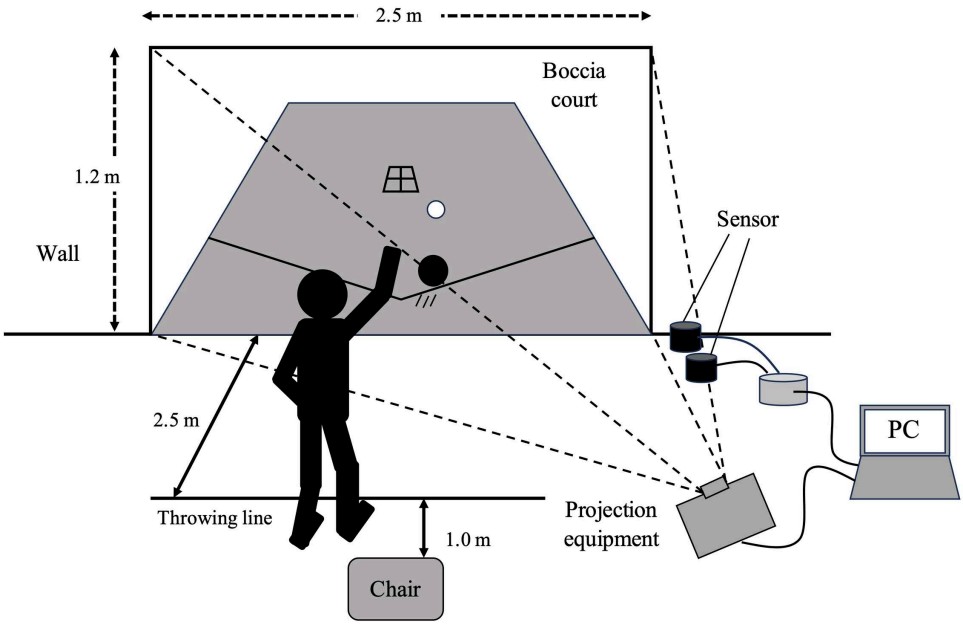

**Fig 2. Experimental set up for Boccia XR software.**

throw relative to previous throws, allowing participants to track game progress. Additionally, when a thrown ball collides with another ball in the projected image, a physics engine simulates the resulting movement based on predefined mechanics. This ensures that ball interactions between follow realistic Boccia dynamics while maintaining an intuitive and engaging experience.

In this study, the participants played Boccia XR in the XR task alongside an examiner. They stood 2.5 m from the wall and threw balls at the projected image. This distance was sufficient to prevent unintended interactions, such as upper limb contact with the wall or unexpected ball rebounds affecting gameplay. No participants reported issues related to proximity to the wall during the experiment.

The examiner initiated the game by throwing the jack ball near the center of the projected court (5 m). The participant procedure was as follows:

(1) Sit on a chair placed 1.0 m from the throwing line.

(2) Stand up and move to the throwing line (2.5 m from the wall) when the prompted.

(3) Throw the ball towards the projected court.

(4) Assume a static standing position, return to the chair, and sit down.

Each participant performed four sets of six throws.

**Boccia.** Participants played traditional Boccia against the examiner. The match began with the examiner throwing the jack ball 5 m away. The procedure was similar to that used for Boccia XR:

(1) Sit on a chair placed 1.0 m from the throwing line.

(2) Stand up and move to the throwing line when instructed.

(3) Throw the ball towards the projected court.

(4) Assume a static standing position, return to the chair, and sit down.

As in the XR task, each participant performed four sets of six throws.

**Treadmill walking.** TR was performed in a controlled laboratory environment under consistent experimental conditions. Participants walked on a level treadmill (Autorunner AR-200; MINATO, Osaka, Japan) at 3.5 km/h for 6 minutes. This speed was set slightly below 3.6 km/h (1.0 m/s), which is commonly regarded as the standard for independent ADL. Since the participants in this study were independently ambulatory, this speed was chosen to ensure feasibility while also accommodating variability in fitness levels, including those whose walking speed may be slightly lower than this standard. The treadmill was gradually accelerated to the target speed at the beginning and decelerated towards the end. Participants were allowed to hold the handrail during speed adjustment but were instructed to avoid using it once the speed stabilized as much as possible.

## Measurements

Demographic data including age, sex, height, weight, and body mass index (BMI), were collected (Table 1). A detailed breakdown of participant characteristics, including individual demographic information, is provided in S1 Table in the Supporting Information. Mood states and lower limb muscle activity were measured to assess the effectiveness of the exercise program.

**Mood state measure.** Mood changes were assessed before and after each task using the short version of the Profile of Mood States 2nd Edition (POMS2) [41,42]. POMS can measure mood changes over short periods [18,43,44] and is considered an appropriate tool for evaluating mood in older adults [41]. POMS2 consists of seven subscales, each related on a 5-point scale (35 items in total), assessing mood states such as Anger-Hostility (AH), Confusion-Bewilderment (CB), Depression-Dejection (DD), Fatigue-Inertia (FI), Tension-Anxiety (TA), Vigor-Activity (VA), and Friendliness (F). The raw scores from these subscales were converted into standardized T-scores, adjust for sex and age. The T-score range in POMS2 typically spans 25 to 92 points. Among these subscales, AH, CB, DD, FI, and TA represent negative mood states, while VA represents a positive mood state. Friendliness (F) is not included in the Total Mood Disturbance (TMD) calculation. The TMD score is computed by summing the negative mood subscale scores (AH + CB + DD + FI + TA) and subtracting the VA score from this total. This method is used because higher VA scores indicate greater vigor and energy, which counterbalance negative mood states. Thus, a lower TMD score reflects an overall improvement in mood. The equation is as follows:

$$TMD\ score\ =\ total\ score\ of\ negative\ mood\ subscales\ \left(AH\ +\ CB\ +\ DD\ +\ FI\ +\ TA\right)\ -\ VA\ score.$$

Enjoyment can be understood as an improvement in mood, where an increase in positive mood and a reduction in negative mood contribute to overall mood enhancement [45]. Therefore, we evaluated the task using VA and TMD scores.

**Table 1. Characteristics of the participants.**

|  | Male (n = 13) | Female (n = 5) | All (n = 18) |
|---|---|---|---|
| **Age (years)** | 73.6 ± 6.1 (65–83) | 72.4 ± 3.1 (67–75) | 73.3 ± 5.4 (65–83) |
| **Height (cm)** | 170.0 ± 7.9 (159.0–183.0) | 155.3 ± 1.9 (153.0–158.0) | 165.9 ± 9.5 (153.0–183.0) |
| **Weight (kg)** | 63.7 ± 7.9 (53.0–81.5) | 50.0 ± 5.8 (40.0–54.0) | 59.9 ± 9.6 (40.0–81.5) |
| **BMI (kg/m²)** | 22.0 ± 2.3 (18.3–26.6) | 20.7 ± 2.2 (16.9–21.9) | 21.7 ± 2.3 (16.9–26.6) |

Mean ± SD (Range), BMI: Body Mass Index.

**EMG data.** EMG data were collected using the Noraxon Ultium system (Noraxon, Scottsdale, USA) from the rectus femoris (RF), semitendinosus muscle (ST), tibialis anterior (TA), and medial gastrocnemius (MG) of the throwing-side lower limb. These muscles were selected for their essential roles in lower limb function, potential stability, and locomotion in older adults [46,47]. They are commonly analyzed in studies on gait and postural control, making them suitable for comparing Boccia XR and treadmill walking [48].

EMG activity was recorded using active electrodes (Blue Sensor M, M-00-s, Mets, Japan), which were applied to lightly abraded skin overeach muscle belly. EMG signals were sampled at 2,000 Hz and bandpass filtered at 80–250 Hz. Recording began before the examiner threw the jack ball and continued throughout the entire task session until all throw were completed.

Participants performed maximum voluntary contractions (MVC) of 3 seconds per muscle, with a rest interval of at least 1 minute between maximal efforts on each experimental day. MVC were performed in standardized postures against manual resistance provided by an examiner, with additional stabilization using non-elastic straps to minimize compensatory movements.

- Rectus femoris: Isometric knee extension in a seated position with 90° hip and knee flexion. The examiner applied resistance at the distal tibia, while both the thigh and pelvis were stabilized with straps to prevent hip movement.

- Semitendinosus: Isometric knee flexion in a prone position with the knee flexed to approximately 60°. The examiner applied resistance at the posterior distal tibia, while the pelvis was stabilized with a strap to prevent excessive hip movement.

- Tibialis anterior: Isometric dorsiflexion in a seated position with the knees at 90° flexion on the edge of a chair or bed. The examiner applied resistance at the dorsal aspect of the foot, while the lower leg was secured with a strap to minimize movement.

- Medial gastrocnemius: Isometric plantarflexion in a long-sitting position on a treatment table with fully extended knees. Participants leaned against a backrest for additional stability, and a non-elastic strap was placed around the pelvis to prevent compensatory trunk movement. The examiner applied resistance at the plantar aspect of the forefoot. A long-sitting position with a backrest was chosen to ensure participant stability during testing.

The recorded MVC values were used to normalize EMG signals (%MVC) for each task. For XR and BO, EMG analysis was performed post hoc, extracting data from the middle eight throws (throws 9–16) to minimize variability due to initial adaptation and potential fatigue effects. The analysis window for each throw began when the participant assumed the throwing stance at the throwing line and ended when they returned to a static standing position after releasing the ball.

%MVC was calculated for each of the middle eight throws (throws 9–16), and the average %MVC was then computed across these throws for each participant. While the exact duration of each throw was not recorded, %MVC normalization ensured meaningful comparisons across conditions despite differences in movement structure.

For treadmill walking, EMG signals were analyzed during the middle 3 minutes of the 6-minute walking period to avoid transient changes in muscle activity at the beginning and end of the trial. A root mean square (RMS) filter (100 ms window) was applied to the EMG signals to smooth the data.

For each task, movement time was normalized to 100% to allow for consistent comparisons across conditions.

**Task duration.** The time required to complete each task was measured to ensure consistency across participants and to account for potential differences in physical exertion among

conditions. The XR and BO tasks began when the examiner threw the jack ball and ended when the participant sat down after the last throw. TR was standardized to 360 seconds. Task duration was not standardized across conditions due to the inherent nature of each activity. While TR had a predefined duration for consistency, XR and BO followed a four-end game format against an examiner. To maintain ecological validity, these tasks were conducted under natural gameplay conditions without imposing an artificial time constraint. Standardizing the duration for Boccia tasks could have altered participants' strategic decision-making, engagement levels, and psychological responses, potentially affecting the effectiveness of the intervention.

## Statistical analyses

All statistical analyses were performed using SPSS ver.28.0 (IBM Corp., Armonk, NY, USA). The normality of the variables was assessed using the Shapiro–Wilk test. For POMS2 data, sphericity assumptions were verified using Mauchly's test, followed by two-way repeated-measures analysis of variance (ANOVA) on VA and TMD scores, with post hoc analysis conducted if the interactions were significant. Bonferroni correction was applied for multiple comparisons, adjusting the significance level to $\alpha = 0.05/3 = 0.0167$ to control for Type I error inflation. EMG data were analyzed using Friedman test, followed by Bonferroni post hoc tests. Statistical significance was set at $p < 0.05$.

To determine the appropriate sample size, an a priori power analysis was conducted using G*Power software. Previous studies on exergame interventions for older adults have reported moderate effect sizes (Cohen's d = 0.54–0.65) in improving balance, physical activity, and psychological well-being [49–51]. However, some studies have raised concerns about usability and acceptance, suggesting that complex interfaces and unfamiliar technology may reduce motivation [52]. In contrast, Boccia XR is designed for intuitive use by older adults, eliminating the need for complex controllers or virtual environments. Its interactive and social nature is expected to enhance motivation and adherence in physical activity interventions. Given these advantages, we hypothesized that Boccia XR would have a greater impact than conventional exergames, potentially leading to a larger effect size. As no direct variance estimates were available from prior research, we adopted Cohen's general guideline for behavioral intervention studies [53]. Based on this guideline, the sample size was calculated for a two-way repeated-measures ANOVA with an alpha level of 0.05, a power (1-β) of 0.80, and an effect size (f) of 0.40, which corresponds to a partial eta squared ($\eta p^2$) of approximately 0.14, classified as a large effect. The power analysis was based on expected differences in VA scores from POMS2, as mood enhancement was a primary outcome of interest. Given prior research on mood enhancement through physical activity, we anticipated a moderate to large effect size in VA scores, justifying the use of an effect size (f) of 0.40 for the analysis. The analysis indicated that a minimum of 12 participants per condition was required to achieve sufficient statistical power. Additionally, for comparisons conducted using the Friedman test, effect sizes were calculated using Wilcoxon's r, following the formula:

$$r = Z / \sqrt{N}$$

The interpretation of Wilcoxon's r followed Cohen's criteria (0.1 = small, 0.3 = medium, 0.5 = large), ensuring consistency in reporting effect sizes across different statistical methods.

## Results

### Mood state measure

Table 2 presents changes in VA and TMD scores of POMS2 before and after each task. Additional details, including the individual scores before and after each exercise condition (XR,

**Table 2. VA and TMD score of POMS2.**

|  | Pre (Mean ± SD) | Post (Mean ± SD) | Interaction task × time | Simple main effect (time) |
|---|---|---|---|---|
| **VA score** | | | | |
| **XR** | 53.7 ± 9.2 | 56.2 ± 9.9 | F = 2.96, p = 0.07 | p = 0.016 |
| **BO** | 54.7 ± 10.9 | 59.1 ± 10.3 | | p < 0.01 |
| **TR** | 56.3 ± 12.2 | 57.3 ± 13.0 | | p = 0.36 |
| **TMD score** | | | | |
| **XR** | 43.6 ± 6.9 | 40.4 ± 5.5 | F = 4.16, p = 0.02 | p < 0.01 |
| **BO** | 42.2 ± 7.0 | 39.7 ± 6.8 | | p = 0.01 |
| **TR** | 41.0 ± 7.5 | 40.5 ± 6.8 | | p = 0.36 |

The results of the two-way repeated-measures ANOVA were significant at p < 0.05.

Effect sizes (ηp², partial eta squared) for ANOVA results are reported in the text.

TMD, Total Mood Disturbance; VA, Vigor-Activity; SD: standard deviation. XR, Boccia XR; BO, Boccia; TR, treadmill walking. Bonferroni correction was applied for multiple comparisons, adjusting the significance level to α = 0.05/ 3 = 0.0167.

BO, and TR), can be found in S2 Table in the Supporting Information. A two-way ANOVA indicated an interaction effect between task and time for VA scores, but it did not reach statistical significance ($F = 2.96$, $p = 0.07$, $\eta p^2 = 0.15$). In contrast, a statistically significant interaction was observed for TMD scores ($F = 4.16$, $p = 0.02$, $\eta p^2 = 0.20$). Simple main effects tests revealed significant differences over time for both VA and TMD scores. Post-hoc tests indicated significant improvements in VA scores after XR and BO tasks ($p = 0.016$, $p < 0.01$), and significant reduction in TMD scores after XR and BO tasks ($p < 0.01$, $p = 0.01$). The partial eta squared values for the simple main effects of time were $\eta p^2 = 0.30$ for XR and $\eta p^2 = 0.45$ for BO in VA scores, while for TMD scores, they were $\eta p^2 = 0.47$ for XR and $\eta p^2 = 0.33$ for BO. No significant improvements were observed in VA and TMD scores for TR (both, $p = 0.36$, $\eta p^2 = 0.05$).

The observed improvements in mood were statistically significant and also suggest clinical relevance, based on the classification thresholds provided in the POMS2 [41]. Post-intervention T-scores for VA and TMD fell within ranges associated with positive mood states, indicating potential psychological benefits. This effect was particularly evident in the XR and BO conditions, where VA scores showed greater increases and TMD scores demonstrated more substantial reductions.

## Muscle activity

Fig 3 presents the mean %MVC of lower limb muscle activity during each task. The raw EMG activation data used to generate this figure is provided in S1 Figure Data in the Supporting Information. The mean %MVC of RF in XR was higher than in TR, but the difference was not statistically significant (Wilcoxon's $r = 0.55$, $p = 0.06$). In contrast, RF activity in BO was significantly higher than in TR (Wilcoxon's $r = 0.75$, $p < 0.01$) (median values: XR 13.3, BO 14.5, TR 8.3%MVC, respectively). The mean %MVC of MG was significantly lower in XR and BO than in TR (XR vs. TR: Wilcoxon's $r = 0.71$, $p < 0.01$; BO vs. TR: Wilcoxon's $r = 0.59$, $p = 0.01$) (median values: XR = 18.4, BO = 18.2, TR = 28.8%MVC). No significant differences were observed in other muscle activities between the tasks. ST activity did not differ significantly among conditions (XR: 14.0, BO: 16.2, TR: 14.9%MVC; XR vs. TR: Wilcoxon's $r = 0.078$, $p > 0.05$; BO vs. TR: Wilcoxon's $r = 0.078$, $p > 0.05$; XR vs. BO: Wilcoxon's $r = 0.16$, $p > 0.05$). Similarly, TA activity did not differ significantly among conditions (XR: 20.2, BO: 21.0, TR: 22.3%MVC; Wilcoxon's $r = 0$ for all comparisons, $p > 0.05$).

## Mean %MVC of Muscle Activity During Each Task

XR: Boccia XR, BO: Boccia, TR: Treadmill Walking    Statistical significance: * p<0.05   ** p<0.01

**Fig 3. The mean muscle activity during each task.** Boxplots representing the %MVC of the rectus femoris, semitendinosus, tibialis anterior, and medial gastrocnemius muscles during task of Boccia XR; XR, Boccia; BO and Treadmill Walking; TR in the participants' throwing side. *p < 0.05, **p < 0.01.

### Task duration

The task completion times for XR and BO were 853.4 ± 17.7 s and 636.5 ± 16.6 s, respectively, while TR was standardized to 360 seconds. Individual task duration data, including start time, end time, and total duration for each participant, are available in S1 File in the Supporting Information. Boccia XR required approximately 14 minute to complete, making it a relatively short-duration activity compared to traditional rehabilitation exercise.

### Discussion

The primary purpose of this study was to determine whether Boccia XR, developed by the authors, meets the essential conditions for exercise in older adults, namely "enjoyment" and "load."

Enjoyment is a crucial determinant of adherence to physical activity programs, particularly among older populations who often struggle to maintain motivation for regular exercise [45]. According to self-determination theory (SDT), intrinsic motivation plays a key role in sustaining physical activity, with autonomy, competence, and social relatedness being primary factors [20]. Boccia XR, by incorporating game-based elements and social interactions, aligns well with these motivational factors, potentially enhancing long-term adherence to exercise in older adults [22]. We hypothesized that Boccia XR and traditional Boccia would result in more positive mood changes than treadmill walking. The POMS2 results showed that both XR and BO increased vigor and decreased negative mood compared to TR. Although the interaction effect for VA scores did not reach statistical significance (p = 0.07), post hoc analyses indicated significant improvements in VA scores after XR and BO tasks. This suggests that Boccia

XR and traditional Boccia may have a positive impact on mood, but further studies with larger sample sizes are needed to confirm this effect. While VA primarily reflects energy and enthusiasm, previous research suggests that positive emotional states, such as increased vigor, are closely associated with the perception of enjoyment in physical activity [10]. Fredrickson's broaden-and-build theory posits that positive emotions enhance engagement and long-term adherence to activities by expanding cognitive and behavioral repertoires [45]. Based on this theoretical framework, our findings suggest that Boccia XR and Boccia, by increasing vigor and reducing negative mood, may contribute to a more enjoyable exercise experience compared to treadmill walking.

MG activity was lower in XR than in TR, while RF activity was higher in BO than in TR. This RF activation is particularly important, as quadriceps strength plays a key role in fall prevention and mobility maintenance [39]. Quadriceps exercises have been shown to improve balance and reduce fall risk in aging populations [52]. Although MG activation was lower in Boccia XR compared to treadmill walking, similar trends have been reported in other task-oriented interventions that emphasize postural control over dynamic propulsion [29]. Overall, muscle activities during XR and BO were comparable to TR, suggesting a similar exercise load and indicating that Boccia can partially function as a lower limb exercise.

Several factors contributed to the positive mood changes observed in XR and BO. First, the social nature of Boccia, which allows multiple participants to engage in the activity together, has been shown to increase exercise motivation among older adults [22,23]. Although our study design did not involve multiple participants playing together, participants engaged in one-on-one gameplay against an examiner. The examiner's role was standardized to maintain consistency across participants; however, even within this structured format, natural interactions occurred during gameplay. Verbal exchanges, encouragement, and the dynamic nature of competition may have contributed to increased motivation and enjoyment. Second, the competitive nature of Boccia, with its clear objectives, has been shown to positively affect the participants' moods [20,21]. Game scores, used as motivational elements, likely contributed to increased vigor. Additionally, the interactive gaming components–especially for Boccia XR–enhanced enjoyment and motivation, similar to the positive effects reported for Wii and VR-based exercises [29–37]. However, individual differences in baseline emotional states and potential order effects may have influenced these findings. Although task order was randomized to minimize biases, residual effects cannot be entirely ruled out. Future studies should incorporate fully counterbalanced designs to further control for these variables and strengthen the validity of the results.

Additionally, many VR-based systems, including Wii Fit and immersive VR training, require head-mounted displays or motion controllers, which may limit accessibility for older adults unfamiliar with such technologies [54–57]. In contrast, Boccia XR retains real-world physical interactions by allowing participants to throw actual balls, ensuring ease of use while still incorporating digital engagement. While this study did not directly compare Boccia XR with other VR-based exercise programs, future research should consider investigating its effectiveness relative to fully immersive virtual exercise interventions. Such comparisons could provide deeper insights into the unique benefits of Boccia XR in enhancing motivation and physical activity adherence among older adults.

Muscle activity varied depending on the task. RF activity was higher during XR and BO than during TR, likely due to slight knee bending during throwing, whereas MG activity was lower. The higher MG activity observed during TR likely reflects the continuous engagement of MG for propulsion and balance. In contrast, Boccia tasks involve intermittent muscle activation patterns, which may explain these differences. Other muscle activities were similar across tasks. The median muscle activity of RF and ST during XR and BO was approximately

15% MVC, while that of TA and MG were ranged from 18% to 22% MVC. Training for muscle strength improvement typically requires muscle activity levels of 60% MVC or higher [58]. However, low-load repetitive training can be effective for muscle hypertrophy and strength improvement even at 25% to 30% 1RM [59–67]. In our study, the muscle activation levels observed in XR and BO (except for TA) were below this threshold required for hypertrophy, indicating that these activities are unlikely to induce significant muscle growth. However, previous studies have shown that even low-intensity resistance exercises can improve neuromuscular function in older adults, particularly those with reduced physical activity [65,66]. While our findings suggest that Boccia XR and traditional Boccia may provide benefits for lower limb function, further studies are needed to determine their long-term impact on muscle conditioning. Given that Boccia XR and traditional Boccia involve repeated lower limb engagement, they may still contribute to muscle endurance, neuromuscular coordination, and postural stability, which are essential for mobility and fall prevention in aging populations. Thus, although Boccia XR and traditional Boccia are not designed as hypertrophy-oriented resistance training, they may serve as effective rehabilitation exercises for maintaining lower limb function and enhancing exercise adherence in older adults.

The XR and BO groups showed similar results in terms of enjoyment and exercise load. Boccia XR can be played in an area one-sixth the size required for traditional Boccia, making it suitable for indoor environments such as hospitals and care facilities. Furthermore, it has the potential to evolve into a system that allows participants to compete remotely via the internet in the future. The time required for each session is approximately 10 to 15 minutes, which has been shown in many studies to be sufficient for mood changes [68–73]. This suggests that the Boccia XR is a realistic and enjoyable exercise program for older adults, offering a practical alternative to longer and more monotonous exercise regimes.

This study had several limitations. The small sample size (n = 18) limits the statistical power and generalizability of the results. Additionally, the study included only healthy older adults, which may limit the external validity of the findings, particularly for populations with different health conditions or functional limitations. Furthermore, none of the participants had prior experience playing Boccia, which may have minimized variability in baseline skill levels but could also have influenced familiarity with the game. Future studies with larger and more diverse populations, including individuals with varying health conditions, are needed to validate these findings and explore the long-term effects of Boccia XR on exercise adherence. Additionally, as each task was performed only once, sustained motivation could not be evaluated. Moreover, throwing posture and intensity were not fully standardized, leading to variability in muscle activity. Finally, compared with treadmill walking, the duration and nature of the Boccia tasks differed. Boccia XR and traditional Boccia followed a natural gameplay format without a fixed duration, whereas treadmill walking was standardized to 360 seconds. While this design maintained ecological validity, differences in task duration may have influenced the results. Future studies should consider time-normalized analyses or standardized task durations to further clarify their impact.

In conclusion, Boccia XR and Boccia may serve as effective lower limb rehabilitation programs, enhancing motivation in older adults. Boccia XR, with gamification elements, is particularly suitable for space-limited environments, as it requires only one-sixth the space of traditional Boccia while maintaining the core gameplay mechanics and engagement factors.

## Supporting information

**S1 Table. Subject characteristics.** Demographic information of all participants, including ID, sex, birth date, age, height, weight, and BMI.
(XLSX)

**S2 Table. POMS scores.** POMS scores before and after each exercise condition: Boccia XR, traditional Boccia, and treadmill walking.
(XLSX)

**S1 Figure Data. EMG activation data for Fig 3.** Raw EMG activation data used to generate Fig 3.
(XLSX)

**S1 File. Task duration data.** Task duration data for each participant in different conditions, including start time, end time, and total duration.
(XLSX)

## Acknowledgments

The authors thank the participants and their families for participating in this study.

## Author contributions

**Conceptualization:** Masataka Kataoka, Kyoji Sugiyama, Akira Iwata, Yumi Higuchi, Ryosuke Saga.

**Data curation:** Masataka Kataoka.

**Formal analysis:** Masataka Kataoka.

**Investigation:** Masataka Kataoka, Kyoji Sugiyama.

**Methodology:** Kyoji Sugiyama, Akira Iwata.

**Project administration:** Akira Iwata, Yumi Higuchi.

**Software:** Ryosuke Saga.

**Supervision:** Akira Iwata, Yumi Higuchi, Shinji Takahashi, Mitsuhiko Ikebuchi, Hiroaki Nakamura.

**Validation:** Masataka Kataoka, Kyoji Sugiyama.

**Visualization:** Masataka Kataoka.

**Writing – original draft:** Masataka Kataoka.

**Writing – review & editing:** Masataka Kataoka.

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
