## [Decision Letter · Decision Letter 0]

30 Dec 2024

PONE-D-24-42948Is Boccia XR an enjoyable and effective rehabilitation exercise for older adults?PLOS ONE

Dear Dr. Kataoka,

Thank you for submitting your manuscript to PLOS ONE. After careful consideration, we feel that it has merit but does not fully meet PLOS ONE’s publication criteria as it currently stands. Therefore, we invite you to submit a revised version of the manuscript that addresses the points raised during the review process.

We look forward to receiving your revised manuscript.

Kind regards,

Hasan Sozen

Academic Editor

PLOS ONE

Journal Requirements:

Reviewers' comments:

Reviewer's Responses to Questions

**Comments to the Author**

1. Is the manuscript technically sound, and do the data support the conclusions?

Reviewer #1: Yes

Reviewer #2: Yes

Reviewer #3: Yes

Reviewer #4: Partly

Reviewer #5: Partly

2. Has the statistical analysis been performed appropriately and rigorously? 

Reviewer #1: Yes

Reviewer #2: Yes

Reviewer #3: Yes

Reviewer #4: No

Reviewer #5: Yes

3. Have the authors made all data underlying the findings in their manuscript fully available?

Reviewer #1: Yes

Reviewer #2: Yes

Reviewer #3: Yes

Reviewer #4: No

Reviewer #5: Yes

4. Is the manuscript presented in an intelligible fashion and written in standard English?

Reviewer #1: Yes

Reviewer #2: Yes

Reviewer #3: Yes

Reviewer #4: Yes

Reviewer #5: Yes

5. Review Comments to the Author

Reviewer #1: The study presents findings from original research and demonstrates its significance. This is a particularly interesting and relevant study within the context of psychological science, especially regarding the adult population, given the numerous factors that justify conducting this and other investigations focused on this specific demographic. The exploration of this population group addresses important gaps in the literature and contributes meaningfully to advancing knowledge in the field.

A notable strength of the present study lies in the breadth and depth of its bibliographic foundation. The bibliographic support provided is significant, highly diversified, and rigorously updated, reflecting a careful and comprehensive engagement with existing literature. This aspect lends substantial credibility to the study and situates it firmly within the broader academic discourse.

Another strength of the study is the depth of its statistical analysis, which is meticulous and well-executed. However, the sample size (N=18) and the participant selection process impose limitations that must be acknowledged. The small sample size, while typical of exploratory research, inherently restricts the ability to establish causal relationships or generalize findings to a broader population. In this context, the study provides valuable insights but cannot definitively confirm causal connections. It is crucial to recognize that a sample size of 18 participants naturally limits the scope for extrapolating results and drawing broader generalizations.

The introduction could be further enriched and expanded to provide a more comprehensive foundation for the study. Specifically, it would benefit from a deeper exploration of the themes and issues directly related to the study’s focus, particularly the challenges faced by the adult population. A more thorough introductory reflection on these challenges, addressing both global and context-specific perspectives, would add significant depth and contextual relevance to the study. Additionally, the introduction would be strengthened by offering clearer and more precise definitions of key concepts and by presenting an overview of the existing psychological science literature relevant to the study. This would help frame the research within a broader theoretical and empirical context.

Moreover, throughout the article, including in the introduction, the quality of the English is generally good but could be further refined. Improving the clarity, fluency, and academic rigor of the language would enhance the overall readability value of the article, thereby increasing its impact and accessibility to a wider audience.

Regarding the methodological section, while it is clear and well-structured, it could be significantly enhanced by providing additional details and justifications. For instance, a more explicit rationale for selecting 18 participants would be valuable. Although this is an exploratory study and therefore does not require a large sample size, a discussion of the implications of the small sample size and its potential limitations would strengthen the methodological rigor. Additionally, comparing the sample size and methodology to similar studies involving the same population type would provide greater context and demonstrate the alignment of the research with established practices in the field.

The study adheres to the ethical standards typical of research in this domain and appropriately complies with established guidelines. However, there are aspects that could be further elaborated upon to enhance transparency and rigor. For example, more detailed explanations of the participant selection criteria, the measures and instruments used, and the procedures implemented to characterize the researchers involved would be valuable additions. Furthermore, outlining the steps taken to minimize bias throughout the research process, as well as providing more specific details on data protection safeguards, would reinforce the ethical robustness of the study. Expanding on the randomization process for participant selection, as well as elaborating on how methodological decisions were aligned with the study’s objectives, would also improve the overall quality of the research.

In conclusion, while the study presents valuable findings and demonstrates methodological strengths, there are opportunities to further refine and enrich the work. Addressing the aforementioned points would enhance its academic value and ensure its contribution to the field is both robust and enduring.

Reviewer #2: 1. Novelty

This study compares Boccia XR with traditional Boccia and treadmill walking, yielding intriguing results. However, have comparisons with other exercise programs for older adults (e.g., Wii Fit, VR-based training) been considered? Virtual reality exercise programs, in general, may be enjoyable and health-promoting. Demonstrating Boccia XR's advantages over these programs would further enhance the value of this research. Even if this is addressed in future studies, please refer to past research on other virtual reality exercise programs and clarify the novelty of this study.

2. Sample and Population

Are the findings of this study applicable to older adults aged 65 and above who are capable of walking? Could there be a bias in recruiting participants already interested in Boccia or exercise? Were participants with prior experience in treadmill walking or Boccia included? Please explain the Population targeted in this study and how the sample was recruited. Furthermore, please elaborate on the rationale for setting a relatively large effect size of 0.40.

3. Boccia XR

The enjoyment of Boccia XR may depend on its user interface. What specific features or innovations are incorporated into Boccia XR? Does it include sound or vibration feedback, as in other video games? Is it designed to prioritize entertainment, making it simpler than traditional Boccia? To ensure the reproducibility of the research, please provide more detailed information about Boccia XR.

Reviewer #3: Dear Authors,

Thank you for valuable work. Looked at the manuscript and here is my decision and recommendations;

Major Revisions

1. Introduction

Clarity and Structure: The introduction sets a clear rationale but could benefit from better alignment between the problem statement, research question, and hypotheses. Explicitly highlight the knowledge gap addressed by the study.

Hypotheses: While the hypotheses are mentioned, they should be more explicitly stated as testable predictions.

2. Methods

Sample Size Justification: The sample size was calculated, but further clarification on the power analysis assumptions (e.g., expected effect size, variance estimates) is needed.

Experimental Design: The order of tasks was randomized, but more details about the randomization method and potential carryover effects should be included.

Task Standardization: Differences in task duration (Boccia XR vs treadmill walking) could confound results. Address why this was not standardized.

3. Results

Statistical Reporting: Statistical results are presented, but effect sizes (e.g., Cohen's d or η²) are missing. Including these will improve result interpretation.

Mood Scores: Highlight whether the observed changes in mood are clinically meaningful, not just statistically significant.

EMG Analysis: Clarify how variability in participant posture during Boccia XR and Boccia might have affected EMG results.

4. Discussion

Interpretation of Findings: The discussion restates results effectively but lacks depth in linking findings to broader literature.

Limitations: While limitations are discussed, potential biases (e.g., participant motivation, novelty effect of Boccia XR) should be expanded.

Future Directions: Suggestions for future research should go beyond standard recommendations (e.g., longitudinal studies, larger samples).

5. Figures and Tables

Figure Clarity: Figures (e.g., EMG data) could benefit from clearer annotations or labels to make results more interpretable.

Table Readability: Ensure consistency in formatting across tables and clarify any ambiguous statistical notations.

6. Ethical Considerations

Conflict of Interest: Explicitly address potential biases due to the involvement of an Institutional Review Board (IRB) chair as a co-author, even with a compliance certificate.

7. Overall Presentation

Language and Grammar: The manuscript is generally well-written but could benefit from minor editing to improve readability and reduce redundancy.

Consistency: Ensure consistent terminology (e.g., Boccia XR vs XR, treadmill walking vs TR) throughout the manuscript.

Over all this is a very important work, congratulations.

Reviewer #4: Dear Editor,

The manuscript presents an innovative approach to rehabilitation for older adults using a gamified Boccia XR system, comparing its effects on mood and muscle activity to traditional Boccia and treadmill walking. While the study addresses a relevant topic and demonstrates novelty, it suffers from some methodological flaws, including a test which measures mood but not specifically enjoyment, a lack of clear methodology for comparing tasks and some questions regarding the statistics.

Given these shortcomings, the manuscript does not meet the scientific and methodological standards required for publication in PLOS ONE. While the concept has potential, extensive revisions are necessary to strengthen the study design, analysis, and interpretation before reconsideration.

Abstract

l.33 Higher compared to what?

l.33 What is sufficient activity?

l.37 I believe you mean that it takes less space than a real boccia court, because it takes up more space than a TM.

Introduction

The introduction explains the problem well. It perhaps lacks some explanation of how they intend on comparing results. For example, it is not clear what you mean by exercise load, nor how it will directly reflect comparable results between classes. The introduction could benefit from deeper contextualisation. For example, why is treadmill walking monotonous? What are other sports used for rehabilitation and how do these compare to boccia?

l.70 You should explain what you mean by exercise load in the introduction.

l.72 What is the goal of analysing muscle activity pre and post activity but not during?

Methods

There is confusion around who played against the subjects in each condition. I have general questions written below and a higher concern for the statistical trend definition. I believe the latter should be removed from the paper and care should be taken when reporting the results. A threshold for statistical significance was given. It is not because a value does not fall within this value that you should report it as something that could be different given other circumstances.

l.80 What was the activity level of your subjects?

l.83 Power analyses are done on specific variables, what variable did you use to justify your sample size?

l.89-l.93 These could be put into one paragraph for more fluid reading.

l.96 at least 3 and at most?

l.96-102 You should make it clearer here who the opponents were. Were subjects put together or sample separately? Wouldn’t this have an effect on your result, as subjects could know each other? Or be more motivated by an opponent who is not “neutral”. For example in line 119-121 you mention that they played with an examiner, who initiated the game by throwing the jack. However, with less able individuals, players can be assisted by one external member. Meaning this examiner could indeed be “with” the subject and assisting them, but not playing against them, which is what you wrote in line 136.

l.119-128 Did the opponent also play?

l.124 Does it matter if they are close to the wall?

l.130 I recommend moving Fig 1 before l.119 as to not place multiple figures together.

l.147 Why this speed? This is also a follow up to the fitness level of your subjects. If they are in good shape, walking at 3.5 km h-1 should be relatively easy.

l.161-178 You should explain this better. In the text you do not explain which parts pertain to the negative mood scales and which parts to the positive mood scales, implying that when the non-initiated reader reaches the equation, they do not understand why you are removing VA for a total score.

l.181-184 Why these muscles?

l.186 What do you consider the task for start and stop? Eight throws? From the beginning of the trial? Over one cycle? More detail should be added for the reader to appreciate the comparison between both sports which are not similar.

l.189 How were MVCs performed?

l.197 Does this mean that in walking you took the averaged activity post-processing over 1 minute and looked at its %MVC? Whereas in Boccia you looked at the average of eight throws. Were the times comparable?

l.191 Why were only 8 throws analysed per condition?

l.211 A trend should not be the same as statistical significance nor infer this (Wood et al., 2014) as stated later in l.216. I suggest removing this clear definition.

Results

l.226 significant tendency.

l.228 what are the values for the Bonferroni correction?

l.242 no statistical difference between XR and TR it should not be specified in a figure.

Discussion

The discussion is to the point and presents arguments for mood and load. It is however missing in some key factors. Subjects only did each session once, limiting some conclusions which can be taken for mood for example. How can you measure for mood on different days without accounting for external factors? What if a person had a bad incident prior to your experiment?

l.261 Please read my comments about multiple participants in the methods and add a part about this here.

l.266 POMS2 measures for positive mood changes and describes VA as a vigorous or energetic measure but not enjoyment. Is this speculation?

l.274 Most muscles shown above (fig 3) did not reach 25-30% of MVC (apart from TA), does this suggest that none of your tasks are sufficient for muscle hypertrophy?

l.282 You present task duration in both the methods and the results section, was the point only to say that this takes around 15 minutes?

Data availability

The PLOS Data policy requires authors to make all data underlying the findings described in their manuscript fully available without restriction, with rare exception (please refer to the Data Availability Statement in the manuscript PDF file). The data should be provided as part of the manuscript or its supporting information, or deposited to a public repository.

l.300 the authors say this is available per reasonable request. This should either be specified as to why they do not want to share per the regulation or publish the data.

Reviewer #5: 1. The study used a sample of 18 healthy elderly participants. For the comparison between two different types of Boccia activities and treadmill walking, this sample size seems small. A small sample may lead to insufficient statistical power. Moreover, the study may be affected by the differences and diversity of the participants, so the universality and reliability of the conclusions are questionable. Besides, short - term experiments cannot verify the impact of Boccia XR on long - term exercise adherence.

2. The study only selected healthy elderly individuals, which may limit the application of the results to elderly people with different health conditions or functional limitations. It is recommended to consider including elderly people with different health conditions to improve the external validity of the results.

3. Although the study mentioned performing different tasks in a random order, it did not elaborate on the randomization method and how to control potential confounding variables, such as the participants' individual differences in responses to different tasks. The lack of detailed randomization and control strategies may lead to bias in the results.

4. The experimental environments of Boccia XR and traditional Boccia are described in relatively detailed ways, but the description of the environment for treadmill walking is insufficient. For example, it is not clear whether the tests were carried out on different days or time periods, which may affect the participants' performance. It is recommended to add relevant explanations.

5. The study reported that Boccia XR and traditional Boccia are significantly better than treadmill walking in enhancing positive emotions. This may be due to the fun brought by social interaction and the nature of games. However, considering individual differences in emotional states and the order effect between tasks, this result needs to be interpreted more cautiously.

6. The study results show that some muscle activity levels of Boccia XR and traditional Boccia are similar to those of treadmill walking. However, there is not enough information to explain why some muscle activities (such as MG) are higher during treadmill walking. A more detailed analysis of the possible reasons for these data differences is needed in the discussion.

6. PLOS authors have the option to publish the peer review history of their article (what does this mean? ). If published, this will include your full peer review and any attached files.

**Do you want your identity to be public for this peer review?** For information about this choice, including consent withdrawal, please see our Privacy Policy .

Reviewer #1: No

Reviewer #2: No

Reviewer #3: **Yes: ** Metin Çınaroğlu

Reviewer #4: No

Reviewer #5: No

---

## [Author Response · Author response to Decision Letter 1]

13 Feb 2025

Dear Editors and Reviewers,

Thank you for your detailed and constructive feedback on our manuscript, Is Boccia XR an enjoyable and effective rehabilitation exercise for older adults? (Manuscript ID: PONE-D-24-42948). We appreciate the time and effort taken to review our work and have carefully addressed all comments.

Major Revisions:

1. Data Corrections:

- We corrected an omission in the POMS analysis, ensuring data from all 18 participants were included.

- We adjusted the EMG analysis window for Rectus Femoris in the XR condition, ensuring accurate interpretation.

- These corrections did not alter the statistical significance of our results.

2. Expanded Introduction and Discussion:

- We provided a more comprehensive background on global and context-specific challenges in aging and physical activity.

- We clarified how Boccia XR differs from other VR-based interventions, emphasizing its accessibility for older adults.

- We integrated Self-Determination Theory (SDT) to explain motivation and adherence benefits.

3. Methodological Improvements:

- We clarified sample size calculations, including power analysis assumptions.

- We detailed the randomization procedure (block randomization) and task balancing.

- We expanded descriptions of Boccia XR’s interface, including real-time visual feedback and physics-based simulation.

4. Expanded Statistical Analysis & Reporting:

- We now include effect sizes (partial eta squared, Wilcoxon’s r) in all key results.

- We addressed concerns about Bonferroni correction in multiple comparisons.

5. Enhanced Figures and Tables:

- We improved figure clarity, revised statistical annotations, and ensured consistency in table formatting.

6. Addressing Reviewer Concerns on Sample & Generalizability:

- We acknowledged the small sample size and self-selection bias and discussed the need for future research on a more diverse population.

- We elaborated on how Boccia XR’s muscle activation levels relate to rehabilitation potential, even though they may not induce hypertrophy.

7. Clarified Ethical Considerations & Data Availability:

- We explicitly stated that the IRB chair did not participate in protocol review to avoid conflicts of interest.

- We ensured compliance with PLOS ONE’s data-sharing policies.

For a detailed point-by-point response, please refer to the attached Response to Reviewers document. We believe these revisions have significantly strengthened our manuscript and appreciate your consideration of our work.

Sincerely,

Masataka Kataoka

Osaka Metropolitan University

---

## [Editor Report · Decision Letter 1]

18 Feb 2025

Is Boccia XR an enjoyable and effective rehabilitation exercise for older adults?

PONE-D-24-42948R1

Dear Dr. Kataoka,

We’re pleased to inform you that your manuscript has been judged scientifically suitable for publication and will be formally accepted for publication once it meets all outstanding technical requirements.

Kind regards,

Hasan Sozen

Academic Editor

PLOS ONE

---

## [Editor Report · Acceptance letter]

PONE-D-24-42948R1

PLOS ONE

Dear Dr. Kataoka,

I'm pleased to inform you that your manuscript has been deemed suitable for publication in PLOS ONE. Congratulations! Your manuscript is now being handed over to our production team.

Kind regards,

on behalf of

Assoc. Prof. Hasan Sozen

Academic Editor

PLOS ONE